# Wheat ABA Receptor TaPYL5 Constitutes a Signaling Module with Its Downstream Partners TaPP2C53/TaSnRK2.1/TaABI1 to Modulate Plant Drought Response

**DOI:** 10.3390/ijms24097969

**Published:** 2023-04-27

**Authors:** Yanyang Zhang, Yingjia Zhao, Xiaoyang Hou, Chenyang Ni, Le Han, Pingping Du, Kai Xiao

**Affiliations:** 1State Key Laboratory of North China Crop Improvement and Regulation, Hebei Agricultural University, Baoding 071001, China; 2College of Agronomy, Hebei Agricultural University, Baoding 071001, China

**Keywords:** *Triticum aestivum*, abscisic acid receptor, drought stress, gene expression, transgene analysis, stomata property, osmolyte content, ROS homeostasis

## Abstract

Abscisic acid receptors (ABR) play crucial roles in transducing the ABA signaling initiated by osmotic stresses, which has a significant impact on plant acclimation to drought by modulating stress-related defensive physiological processes. We characterized *TaPYL5*, a member of the ABR family in wheat (*Triticum aestivum*), as a mediator of drought stress adaptation in plants. The signals derived from the fusion of TaPYL5-GFP suggest that the TaPYL5 protein was directed to various subcellular locations, namely stomata, plasma membrane, and nucleus. Drought stress significantly upregulated the *TaPYL5* transcripts in roots and leaves. The biological roles of ABA and drought responsive *cis*-elements, specifically ABRE and recognition sites MYB, in mediating gene transcription under drought conditions were confirmed by histochemical GUS staining analysis for plants harbouring a truncated *TaPYL5* promoter. Yeast two-hybrid and BiFC assays indicated that TaPYL5 interacted with TaPP2C53, a clade A member of phosphatase (PP2C), and the latter with TaSnRK2.1, a kinase member of the SnRK2 family, implying the formation of an ABA core signaling module TaPYL5/TaPP2C53/TaSnRK2.1. TaABI1, an ABA responsive transcription factor, proved to be a component of the ABA signaling pathway, as evidenced by its interaction with TaSnRK2.1. Transgene analysis of *TaPYL5* and its module partners, as well as *TaABI1*, revealed that they have an effect on plant drought responses. *TaPYL5* and *TaSnRK2.1* positively regulated plant drought acclimation, whereas *TaPP2C53* and *TaABI1* negatively regulated it. This coincided with the osmotic stress-related physiology shown in their transgenic lines, such as stomata movement, osmolytes biosynthesis, and antioxidant enzyme function. *TaPYL5* significantly altered the transcription of numerous genes involved in biological processes related to drought defense. Our findings suggest that *TaPYL5* is one of the most important regulators in plant drought tolerance and a valuable target for engineering drought-tolerant cultivars in wheat.

## 1. Introduction

Drought is one of the most damaging abiotic stresses, having a significant impact on plant growth and crop productivity [1,2]. Crop production has become water-limited in recent years and will be further worsened in the future as a result of global climate change [3,4]. In such cases, the ability of plants to survive in drought conditions is critical for crop growth and development [5,6]. As a result, extensive research on responses of crops to water deficit has become a hotspot for engineering crop cultivars to be high-yielding and drought-tolerant.

Plants have evolved a number of cellular and molecular signaling pathways to activate and regulate plant resistance to osmotic stresses. Numerous studies have been conducted to evaluate the mechanisms underlying plant drought response at the morphological, physiological, and biochemical levels [7,8]. Drought stresses alter root growth characteristics such as root length, diameter, surface area, tissue density, biomass, and branch number [9,10,11], which are crucial for plants to maintain appropriate biological functions in a more humid environment and sustain water uptake under adverse environmental conditions [12,13]. Furthermore, osmotic adjustment is a major strategy of plants in response to drought by regulating plant-water relations and maintaining plant growth by regulating root cell expansion [14,15,16]. To avoid damage from reactive oxygen species (ROS) initiated by drought stress, plants use a variety of antioxidants, including enzymes [17,18] and non-enzymatic metabolites [19].

Abscisic acid (ABA) is a key mediator in plant response to drought stress [20] and an important regulator in modulating plant growth, development, and responses to a variety of abiotic stresses [21]. Under drought conditions, stomatal movement is mediated largely by the ABA molecule to inhibit plant water loss from transpiration [22]. Over the last two decades, it has been established that the ABA-dependent signaling pathway is essential in transducing the ABA signaling elicited by osmotic stress and functions as a central modulator in mediating plant drought response. Among others, ABA receptors (ABR) such as PYRABACTIN RESISTANCE (PYR) or regulatory components of ABA receptors (RCAR), have been shown to improve ABA responses and confer drought tolerance in the model plant *Arabidopsis thaliana* [23]. Fourteen members of such proteins have been identified in *A. thaliana*, namely pyrabactin resistance 1 (PYR1) and PYR1-like 1-13 (PYL1-PYL13) [24], or the regulatory component of ABA receptor (RCAR1-RCAR14) [25]. Based on crystallographic studies, a well-accepted mechanism for ABA recognition by PYL proteins was discovered. ABA signals are perceived and transduced via a gate-latch-lock mechanism. ABA binds to the PYL receptor’s ligand binding pocket to induce a conformational change leading to pocket closure via the gate and latch structure and creating an interaction surface for the downstream clade A phosphatase (PP2Cs) [26]. The ABA-induced interaction of PYL proteins with PP2C members releases PP2C’s inhibitory effect on kinase activation of SnRK2 family members, resulting in the activation of SnRK2 proteins, which further phosphorylates downstream targets such as ABF transcription factors [27]. These findings suggested that although the signaling components constituting the ABA signaling pathway are inactive under non-stress conditions [27], they are activated upon drought and contribute greatly to the signaling transduction as well as plant stress response. Further characterization of the molecular processes by which ABA receptors bind the ABA molecule elicited by drought can aid in the rational design of bioactive molecules to improve drought resistance in crop cultivars [28,29].

Despite a large number of studies focusing on functional characterization of the ABA signaling pathways having been conducted in the model plant *Arabidopsis*, the ABA signaling pathway, such as identification of members constituting the ABA core signaling module, namely, PYL/PP2C/SnRK2, as well as its downstream regulators involving transducing osmotic stress-initiated ABA signals has been poorly dissected. The determination of their biological roles in cereal crops under drought conditions is still limited. In this study, we reported *TaPYL5*, an ABR member in wheat (*Triticum aestivum* L.), focusing on elucidating its drought response at the transcriptional level, identifying its partners to establish an ABA core signaling network, and characterizing their roles in mediating plant response to drought stress. Our findings demonstrate that *TaPYL5* is drought sensitive. The TaPYL5 protein, along with its PP2C and SnRK2 family partners, TaPP2C53 and TaSnRK2.1, and an ABA responsive transcription factor TaABI1, are involved in the establishment of an ABA signaling pathway that modulates plant drought response.

## 2. Results

### 2.1. Molecular Characterization of TaPYL5

The full-length cDNA of *TaPYL5* (TraesCS1A02G126800) is 606 bp and encodes a 201-aa polypeptide with a molecular mass of 22.16 kDa and an isoelectric point (pI) of 5.79. The TaPYL5 protein shares nine conserved domains (i.e., CL1 to CL9) with its counterparts from other plant species, all of which are involved in binding ABA and interacting with downstream proteins such as members of the PYL family (Figure 1A). At the nucleic acid level, *TaPYL5* shares a high identity with ABR genes found in a variety of plant species, particularly PYL genes found *Aegilops tauschii PYL5* (99.76%, XM 020339365), *Festuca elata PYL5* (94.44%, KY475601), *Hordeum vulgare PYL5* (98.07%, XM 045112029), *Panicum virgatum PYL5* (88.81%, XM039980237), and *P. hallii PYL5* (88.81%, XM 025952729), etc. (Appendix A). The high sequence similarity to ABR genes in plant species suggests that *TaPYL5* functions as a member of the PYL family in wheat. An experiment using the model plant *Nicotiana benthamiana* system was used to determine the sub-cellular position of the fusion TaPYL5-GFP. GFP signals from the fusion in epidermal cells of the fusion-carrying model plant system TaPYL5-GFP was shown to be restricted in stomata, plasma membrane, and nucleus (Figure 1B). These findings indicate that the TaPYL5 protein targets the above positions after being sorted from the endoplasmic reticulum (ER), where it performs related biological functions.

### 2.2. TaPYL5 Expression Is Sensitive in Response to Drought Stress

*TaPYL5* transcripts were examined during drought treatments to characterize its response to this type of stressor. The *TaPYL5* expression levels were significantly upregulated in both roots and aerial tissues in response to drought signaling, displaying a pattern of gradually increasing transcript abundance along with the intensification of drought extent initiated by polyethylene glycol (PEG-6000) with concentrations ranging from 0 to 15% (*w/v*)) (Figure 2A). Furthermore, analysis of the *TaPYL5* temporal expression pattern revealed that its transcripts in both tissues mentioned were elevated within a 27-h drought treatment regime, and the drought-induced transcripts were restored following progression of a 27-h normal recovery condition (Figure 2B). These findings suggest that the *TaPYL5* transcription is sensitive to drought stress.

### 2.3. A Suite of Cis-Regulatory Elements in TaPYL5 Promoter Impacts Gene Responses to Drought Stress

A prediction online tool (PLACE) was used to identify the *cis*-acting regulatory elements located in the *TaPYL5* promoter, 2-kb in length upstream of the translation start codon (ATG). The *TaPYL5* promoter was found to contain a subset of critical elements, including those involved in RNA polymerase interaction (TATA box), gene expression efficiency regulation (CAAT box), and osmotic stress signaling response (i.e., ABRE and DRE elements) (Figure 2C). Furthermore, the *TaPYL5* promoter contains a subset of binding sites for transcription factor MYB proteins, such as MYB88, MYB46, and MYB14, which are frequently involved in gene transcription in response to osmotic stress (Figure 2D). As a result, these *cis*-acting regulatory elements may influence the *TaPYL5* response to drought at the transcriptional level.

To determine the roles of critical *cis*-elements in mediating gene transcription efficiency under drought conditions, histochemical GUS staining assays were performed. The results revealed that once driven by truncated promoter fragments lacking distinct elements/element amounts, the expression levels of the GUS reporter were altered. When compared to the staining extent in the control, which was integrated by a full-length (1938 bp) promoter, the GUS staining extent in samples integrated by a set of truncated promoter regions was drastically reduced. This indicates that the GUS staining extent is closely correlated with the promoter length integrated under drought conditions. Although reduced GUS staining extents were detected in samples driven by truncated promoter regions, all of the staining extents in leaf samples harboring *TaPYL5*pro378-*GUS* (D1, −378), *TaPYL5*pro640-*GUS* (D2, −640), *TaPYL5*pro1055-*GUS* (D3, −1055), *TaPYL5*pro 1635-*GUS* (D4, −1635), and *TaPYL5*pro1938-*GUS* (D5, −1938) treated by drought displayed stronger GUS staining and activities than the control (D5) cultured under normal growth conditions (Figure 2E). These results suggest that the *cis*-acting regulatory elements located at various positions in the promoter, such as elements ABRE and DRE and recognition sites of MYB, exert positive functions in mediating the transcriptional response of *TaPYL5* to drought stress.

### 2.4. TaPYL5 Involves the Constitution of an ABA Core Signaling Module with PP2C and SnRK2 Members

The TaPYL5-involved establishment of an ABA core organelle module was determined by yeast two-hybrid and BiFC assays. The yeast two-hybrid assay revealed that the TaPYL5 protein interacted specifically with a PP2C member known as TaPP2C53 (Figure 3A). The TaPP2C53 protein specifically interacted with TaSnRK2.1, a member of the SnRK2 family in wheat, according to further analysis using a similar strategy for identifying TaPYL5-TaPP2C53 protein interaction (Figure 3B). Therefore, this suggests that TaPYL5 establishes an ABA core signaling module with its downstream partners, namely TaPYL5-TaPP2C53-TaSnRK2.1 that is thought to transduce drought-induced ABA signaling and to mediate plant drought response.

The BiFC analysis confirmed the protein-protein interactions observed in the yeast two-hybrid assay. As expected, strong YFP signals derived from TaPYL5-nYFP, TaPP2C53-cYFP, TaPP2C53-cYFP and TaSnRK2.1-nYFP were detected in leaves of the *Nicotiana benthamiana* model plant system co-transformed with each of the above cassette combinations (Figure 3C). In summary, TaPYL5 participates in the establishment of an ABA core signaling module with its downstream partners, which transduces the ABA signaling and mediates the plant response to drought stress by regulating the related physiological processes.

### 2.5. Transgenic Lines of Genes Encoding TaPYL5 and Its Downstream Partners Modify Plant Drought Response

*TaPYL5* and its downstream partner genes were subjected to transgene analyses in wheat in order to characterize their functions in response to drought stress. The transgenic lines of *TaPYL5* (Sen 2, Sen 3, Anti 1, and Anti 2), *TaPP2C53* (Sen 1, Sen 2, Anti 1, and Anti 2), and *TaSnRK2.1* (Sen 3, Sen 4, Anti 2, and Anti 3) with typical overexpression or knockdown of the target (Appendix A), were subjected to normal growth and drought treatments. Under normal growth conditions, all of these transgenic lines exhibited similar phenotypic behaviors, tissue biomass both aboveground and root, and root volume with the wild-type (WT) (Figure 4 and Figure 5A–D). Under drought treatments, the transgenic lines were modified for the above growth traits, with improved growth and increased biomass of aboveground and root tissues, as well as elevated root volumes, observed in lines overexpressing *TaPYL5* (Sen 2 and Sen 3), and inhibited growth, decreased biomass, and lowered root volumes observed in lines with *TaPYL5* knockdown expression (Anti 1 and Anti 2) compared to the WT (Figure 4 and Figure 5A–D). The lines with overexpression or knockdown expression of *TaSnRK2.1* were similar to *TaPYL5* lines in terms of the traits assessed, whereas *TaPP2C53* lines were reversed in terms of the traits assessed (Figure 4 and Figure 5A–D). The transgene analyses confirmed the function of this ABA core signaling module, specifically the positive roles of *TaPYL5* and *TaSnRK2.1* and the negative role of *TaPP2C53* in regulating plant drought response.

### 2.6. TaPYL5-Mediated Drought Response Is Associated with the Role in Modulating Stomata Movement, Osmolyte Accumulation, and ROS Homeostasis

The osmotic stress-related physiological indices, such as stomata closing rate (SCR), leaf water losing rate (WSR), proline and soluble sugar contents, and ROS homeostasis parameters, were analyzed for the *TaPYL5*, *TaPP2C53*, and *TaSnRK2.1* lines. In accordance with the plant growth traits, the *TaPYL5* lines (Sen 2, Sen 3, Anti 1, and Anti 2) were found to be similar on the stress-related indices tested, demonstrating comparable SCR in a 2 h drought treatment regime, WSR in a 3 h water deficit condition. Similarly, under normal conditions, the contents of proline and soluble sugar, as well as the activities of antioxidant enzymes superoxide dismutase (SOD), peroxidase (POD), catalase (CAT), and malondialdehyde (MDA), were compared to the WT. Under drought conditions, the *TaPYL5*-overexpressing lines (Sen 2 and Sen 3) exhibited improved SCR (Figure 6A,B), decreased WLR (Figure 6C), increased osmolytes (proline and soluble sugar) (Figure 6D,E), enhanced SOD, POD, and CAT activities (Figure 6F–H), and decreased MDA accumulation (Figure 6I). In contrast to the preceding lines, those with *TaPYL5* knockdown expression (Anti 1 and Anti 2) demonstrated the opposite behavior on those indices (Figure 6A–I). The results of the above osmotic stress-related physiological indices of *TaPP2C53* (Sen 1, Sen 2, Anti 1, and Anti 2) and *TaSnRK2.1* (Sen 3, Sen 4, Anti 2 and Anti 3) lines were also consistent with their growth characteristics. According to the physiological indices obtained from transgene analyses, the *TaPYL5*-involved establishment of an ABA core organelle module mediates plant drought tolerance, which is closely associated with the modulated stress responsive-associated physiological processes.

### 2.7. TaSnRK2.1 Protein Interacts with ABA-Responsive Transcription Factor TaABI1 to Negatively Impact Plant Drought Resistance

To better understand the molecular process underlying the ABA signaling module-mediated plant drought response, the putative downstream partner(s) of TaSnRK2.1 were identified using a yeast two-hybrid assay. TaABI1, an ABA responsive-associated transcription factor, was found to interact with TaSnRK2.1 specifically (Figure 7A). Further BiFC assays confirmed the interaction between TaSnRK2.1 and TaABI1; strong YFP signals were detected in *Nicotiana benthamiana* explants co-transformed by *TaSnRK2.1-nYFP* and *TaABI1-cYFP* (Figure 7B). These findings suggested that TaABI1 functions as a downstream partner in the TaPYL5-involved module regulation, potentially contributing to plant drought response.

Transgenic lines with modified *TaABI1* expression were created in order to better understand its role in mediating plant drought response. Lines OE 1 and OE 2, two wheat *TaABI1*-overexpressing lines, and KE 1 and KE 2, two wheat *TaABI1* knockdown lines that were specified by target overexpression or knockdown expression (Appendix A), were subjected to drought treatment. Results showed that they were heavily influenced by growth traits and osmotic stress-related physiological indices, such as SCR, WSR, and contents of oymolytes. The plant growth was inhibited and the related physiological indices (i.e., decreased SCR, increased WLR, and proline and soluble sugar contents) were decreased in lines OE 1 and OE 2, whereas KE 1 and KE 2 had the opposite effect on the traits shown in the above lines with *TaABI1* overexpression (Figure 7C–H). These findings suggest that TaABI1 acts as a regulator downstream of TaSnRK2.1 to negatively regulate plant drought adaptation.

### 2.8. Transcriptome Profile Mediated by TaPYL5 upon Drought Signaling

The effects of *TaPYL5* on the transcriptome were investigated using an RNA-seq analysis on a drought-stressed *TaPYL5*-overexpressing line (Sen 1) and the WT plants as samples. The results revealed that 1669 genes were differentially expressed in Sen 1 compared to the WT, with 610 upregulated and 1059 downregulated (Appendix A). To verify the reproducibility of transcriptome results, qRT-PCR was performed on ten differentially expressed genes (DEGs), five of each of which were upregulated or downregulated in expression. The results on transcripts detected from all of them were consistent with those shown in the RNA-seq analysis, namely, the expression levels of the five upregulated genes (*TaPT5, TaLD2, TaWRKY9, TaNPP,* and *TaF3H*) were increased, whereas those of the other five DEGs with downregulated expression (*TaWAT1, TaADH2D, TaSADP, TaCBF,* and *TaH2B*) were decreased in Sen 1 with respect to the WT (Appendix A). The qRT-PCR results for these DEGs validated the transcriptome pattern underlying *TaPYL5* modulation in response to drought stress.

Gene ontology (GO) analysis on the DEGs classified them into the GO terms associated with ‘biological process’, ‘molecular functions’, and ‘cellular component’. Thirteen subgroups of DEGs are overrepresented by functions associated with ‘biological process’, including cellular processes and metabolic processes, among others. The DEGs in the term ‘molecular function’ include seven subgroups that are functional in molecule binding, modulating catalytic activity of enzymes, and regulating transcription regulator activity, etc. The DEGs in the term ‘cellular component’ include 12 subgroups that are effective in affecting cell part, organelle, and membrane part, among other things (Figure 8A). The DEG-modulated biochemical pathways were further defined by the KEGG analysis. Over 50 pathways underlying *TaPYL5* modulation are proposed to have been altered, particularly those associated with ion trafficking and homeostasis, trivalent inorganic anion homeostasis regulation, and glucosamine-containing compound metabolism, among others (Figure 8B). These findings suggest that *TaPYL5* has a broad impact on regulating plant drought response by modifying gene transcription at the global level.

## 3. Discussion

### 3.1. TaPYL5 Response to Drought Signaling Associates with a Set of Cis-Elements Situated in Promoter

While exposed to stressors, the transcription efficiencies of genes in the ABA signaling pathway are altered [30,31,32]. Investigations into members of the ABA receptor family have documented their responses to osmotic stress at the transcriptional level. For example, expression analysis of *AtPYL*, a PYR member of *A. thaliana*, revealed that it was upregulated on the transcript abundance under drought and high salinity conditions, which led to altered transcription of a subset of stress defensive-associated genes and improved plant acclimation to the above stressors [33]. Our analysis of *TaPYL5* expression patterns during drought signaling revealed that it is sensitive to drought stress; more *TaPYL5* transcripts were detected in roots and aerial tissues under intensified stress extent and the progress of stress duration. This finding suggests that it plays a role in transducing drought signaling, impacting on plant stress response. So far, a set of *cis*-acting regulatory elements in promoters that modulate plant response to ABA and drought signaling has been identified. Of these, the elements ABRE (specified by motifs ACGTG, AACCCGG, and CGTACGTGCA) [34] and the transcription factor recognition and binding sites, such as MYB, bZIP, and MYC members, all contribute to gene response to drought stress at the transcriptional level [35]. We identified the *cis*-acting elements responsive to ABA and drought through an online search of the *TaPYL5* promoter. Further GUS histochemical staining assays to define reporter expression under control of the *TaPYL5* promoter fragments, which included or lacked the aforementioned distinct *cis*-elements, confirmed their positive roles in regulating gene transcription efficiency. Further characterization of these *cis*-acting elements using base mutation technology can help better understand the mechanism of *TaPYL5’*s drought response.

### 3.2. TaPYL5 Is Involved in Establishing an ABA Signaling Module with Distinct PP2C and SnRK2 Members to Contribute to Plant Drought Response

The ABA core signaling module, which is made up of different ABR members and its downstream partners, plays critical roles in signaling transduction, and its dissection aids greatly in understanding the mechanism of module-mediated plant acclimation to drought stress [36]. Our previous investigation into TaPYL4, a member of the ABA receptor family, indicated that it is involved in the constitution of an ABA core signaling module with PP2C member TaPP2C2 and SnRK2 member TaSnRK2.1 [37]. This study was initiated to better understand the ABA core signaling module composed of a TaPYL5 protein and PP2C member, as well as the PP2C protein interacting protein(s) in the SnRK2 family. The yeast two-hybrid and BiFC assays were used to identify a PP2C member, TaPP2C53, and a SnRK2 protein, TaPP2C.1, both of which are involved in the establishment of an ABA core signaling module. As a result, this module TaPYL5/TaPP2C53/TaSnRK2.1 is proposed to be functional in transducing drought-induced signaling in wheat. ABR and its downstream partners’ interaction and action mechanisms were previously defined [38]. The ABR protein, for example, changes its conformation and then binds ABA induced by osmotic stresses, and the mobile gating loop established after ABA binding allows it to interact with a distinct clade A PP2C protein by interacting with the conserved active site [38,39]. Our protein analysis on the three-dimensional structure revealed conserved motifs in the TaPYL5 protein, implying that they are potentially involved in the protein-protein interaction process. However, detailed molecular mechanisms of how the TaPYL5 protein interacts with its partner, TaPP2C53, need to be elucidated. Further research on the dynamic molecule coupling between TaPYL5 and TaPP2C53, as well as the latter and its downstream partner, TaSnRK2.1, can help us understand the action mechanism of the ABA core signaling component in wheat. However, the detailed molecular mechanisms by which the TaPYL5 protein interacts with its partner, TaPP2C53, need to be determined.

### 3.3. TaPYL5 and Genes Encoding Its Partners Contribute Largely to Plant Drought Response

Functional characterization of ABR family members has revealed that they are extensively involved in mediating diverse physiological processes in plants, such as seed dormancy, germination, lateral root formation, light signaling convergence, and flowering time control [40,41,42]. Further investigations have also confirmed their roles in promoting plant acclimation to osmotic stresses. For example, studies into *AtPYL5*, *AtPYL8*, and *AtPYL9*, three members of the PYL family in *A. thaliana*, revealed that they act as crucial regulators in mediating plant response to exogenous ABA during seed germination and seedling growth, promoting stomatal movement and improving plant stress resistance [43,44,45]. A number of the genes of the PYL family including *OsPYL3*, *OsPYL5*, and *OsPYL9* in rice (*Oryza sativa*), *GhPYL10*, *GhPYL12* and *GhPYL26* in cotton (*Gossypium herbaceum*), and *ZmPYL3*, *ZmPYL9*, *ZmPYL10* and *ZPYL13* in maize (*Zea mays*), enhance the plant response to ABA under osmotic stress conditions, positively modulating osmolyte accumulation, and improving plant biomass production ability [46,47,48]. In wheat, previous investigations into *TaPYL4* have indicated that it acts as one of the critical regulators in positively regulating plant drought tolerance [37,49]. The transgenic wheat lines with *TaPYL4* overexpression displayed increased plant ABA sensitivity, lowered water consumption during the plant’s lifetime by reducing transpiration, increased photosynthetic activity, and improved water-use efficiency under water deficit conditions [49]. To characterize the functions of *TaPYL5*, *TaPP2C53*, and *TaSnRK2.1* in mediating plant drought response, we generated transgenic wheat lines for *TaPYL5*, *TaPP2C53*, and *TaSnRK2.1*. The lines for target genes with overexpression or knockdown expression showed significant changes in growth behaviors, such as phenotypes and plant biomass. *TaPYL5* and *TaSnRK2.1* positively and *TaPP2C53* negatively regulated plant adaptation to drought stress. These results from *TaPYL5* and its partners confirmed the importance of the TaPYL5/TaPP2C53/TaSnRK2.1 action module in transducing drought signaling via an ABA-dependent pathway and regulating plant drought response.

Drought adaptation of plants is closely related to stress-related physiological processes, such as stomata movement that is adapted to closing to avoid water loss by transpiration [50,51], induction of osmotic substance accumulation [52], and enhancement of ROS homeostasis by scavenging ROS via antioxidant enzyme capacity enhancement [53,54]. Our investigation into *TaPYL4* revealed that this wheat-ABA-receptor gene-mediated plant drought response is associated with its role in transcriptionally regulating *TaPIN9,* a PIN-FORMED gene modulating cellular auxin translocation. The modified expression of this PIN-FORMED member improves the establishment of root system architecture (RSA) underlying *TaPYL4* regulation to contribute to plant drought adaptation [37]. In this study, we investigated the physiological parameters related to plant osmotic stress response to better understand plant drought responses mediated by the *TaPYL5*-involved ABA signaling pathway. Lines overexpressing *TaPYL5* and *TaSnRK2.1* showed increased SCR when challenged by drought signaling, increased water retention capacity in drought-treated detached leaves, elevated osmolytes (i.e., proline and soluble sugar contents), and increased AE activities (i.e., SOD, CAT, and POD). Lines with *TaPP2C53* overexpression, on the other hand, displayed opposite patterns on the above indices to those seen in the *TaPYL5* and *TaSnRK2.1* lines. The plant growth behaviors underlying *TaPYL5* modulation and its partners constituting an ABA core signaling module are consistent with the protein-protein interaction mechanism involved in drought-initiated signaling. As a result, the signaling module composed of PYL, PP2C, and SnRK2 members in wheat, namely TaPYL5/TaPP2C53/TaSnRK2.1, is functional in plant drought acclimation by its large impact on stress-related physiological processes.

### 3.4. TaABI1 Is a Downstream Component of TaPYL5 Signaling Module and Negatively Regulates Drought Adaptation

Members of the ABA responsive transcription factor family (ABI) modulate gene transcription in response to exogenous ABA and act as a critical regulator in plant osmotic stress response, affecting a subset of stress-responsive-related physiological processes [55]. We investigated whether any member of the TaABI family connects the TaPYL5-involved module in plant drought response by analyzing putative protein-protein interactions between TaSnRK2.1 and a member(s) of the TaABI family. The yeast two-hybrid and BiFC protein-protein interaction assays revealed that TaABI1 specifically interacted with TaSnRK2.1. This finding suggests that TaABI1 may be involved in drought signaling transduction via the ABA core signaling pathway established by TaPYL5 and its partners. The transgenic wheat lines of *TaABI1* showed modified growth behaviors (phenotypes and plant biomass), altered physiological indices such as stomata movement and osmolytes (proline and soluble sugar contents) with respect to the WT under drought treatment; plant drought adaptation was reduced in lines with *TaABI1* overexpression, whereas plant drought tolerance was improved in lines with gene knockdown. These findings establish a link between the *TaPYL5*-involved ABA signaling module and its downstream partner TaABI members, which build the TaPYL5/TaPP2C53/TaSnRK2.1-TaABI1 signaling pathway to act cooperatively in regulating plant drought response.

### 3.5. TaPYL5 Modifies Gene Transcription Globally to Impact Plant Drought Response-Associated Physiological Processes

It has been shown that ABA signaling induced by osmotic stressors causes a significant change in the expression of a number of genes involved in mediating plant osmotic stress response, thereby altering stress-related physiological parameters and biochemical pathways [56,57]. To gain a better understanding of the molecular processes underlying the *TaPYL5*-mediated plant drought response, we used RNA-seq to identify the genes with altered transcription efficiency under drought conditions. A large number of genes on transcripts in the transgenic line with *TaPYL5* overexpression (Sen 1) were significantly altered when compared to the drought stressed WT plants. According to KEGG, they regulate over fifty biochemical pathways, with a focus on ion trafficking and homeostasis, trivalent inorganic anion homeostasis regulation, and glucosamine-containing compound metabolism. The transcriptome profile analysis suggests that *TaPYL5* has a broad impact on modulating gene transcription that regulates plant acclimation/adaptation to drought stress. Based on our findings, we proposed a model for *TaPYL5*, an ABR member in wheat, to mediate plant drought response (Figure 9). *TaPYL5* transcripts are induced in plant roots and aerial tissues in response to drought signaling. After binding ABA molecules, the induced TaPYL5 proteins acclimatize to interact with TaPP2C, a member of the clade A phosphatase (PP2C), via a protein-protein interaction mechanism. The formation of this trimer complex molecule (TaPYL5/ABA/TaPP2C53) eliminates TaPP2C53’s inhibition of TaSnRK2.1, a member of the SnRK2 family in wheat. Then, TaSnRK2.1 interacts with TaABI1, an ABA-responsive TF in wheat that extensively modulates transcription of genes involving regulation of osmotic stress-associated physiology, such as stomata movement, osmolytes biosynthesis, and cellular ROS homeostasis, to contribute to plant adaptation to drought stress. Further identification and functional dissection of unknown components acting as members of the *TaPYL5*-involved signaling pathway may provide insights into the mechanism of plant osmotic stress resistance.

## 4. Materials and Methods

### 4.1. Characterization Analysis on TaPYL5

*TaPYL5* (GenBank accession No. TraesCS1A02G126800), a gene in wheat with a high nucleic acid sequence similarity to ABA receptor (PYL) members in our previous RNA-seq analysis (unpublished data), was found to be significantly upregulated in expression during drought stress. The open read frame (ORF) and a 2 kb promoter region of *TaPYL5* situated on chromosome 1D (*TaPYL5-1D*, hereafter named *TaPYL5* for simplicity) and those of its homeologs located on chromosomes 1A (*TaPYL5-1A*, TraesCS1B02G145800) and 1B (*TaPYL5-1B*, TraesCS1D02G126900) were obtained from Ensembl Plants (http://plants.ensembl.org/index.html accessed on 22 April 2023), which have a high degree of similarity at the nucleic acid and amino acid levels (98.18–98.51%, Appendix A). *TaPYL5*, in comparison to other homeologs, showed significantly induced transcripts in response to drought stress (Appendix A). Conserved domains of TaPYL5 protein were defined by plant PYL counterparts as previously described [58]. Phylogenetic tree covering *TaPYL5* and its homologous genes across various plant species were established using the MegAlign algorithm supplemented in DNAStar software (www.dnastar.org accessed on 22 April 2023). The sequences used in phylogenetic relation analysis were obtained with a BLASTn search against the National Center for Biotechnology Information’s (NCBI) GenBank database (www.https://blast.ncbi.nlm.nih.gov/Blast.cgi accessed on 12 January 2022).

### 4.2. Analysis of TaPYL5 Protein Localization at the Subcellular Level

An experiment was carried out in order to determine the subcellular localization of the TaPYL5 protein after endoplasmic reticulum (ER) assortment. The ORF of *TaPYL5* was amplified using gene specific primers (Appendix A) using a reverse transcription-polymerase chain reaction (RT-PCR) and then integrated into the binary vector pCAMBIA3300 in a frame with the reporter gene, a green-fluorescent-protein encoding gene (*GFP*), downstream of the CaMV35S promoter. The expression cassette *TaPYL5-GFP* was genetically transformed onto the *Agrobacterium tumefaciens* strain EHA105 using the standard heat-shock method. Following the *A. tumefaciens*-mediated transformation method, the transformants containing the cassette were used to genetically transform epidermal cells of *N. benthamiana*. After 48 h of infiltration, the GFP signals initiated by the fusion of TaPYL5-GFP, as well as those initiated by the control CaMV35S-GFP integrated by an empty vector, were detected under a fluorescent microscope, as previously described [59]. Furthermore, a protoplast expression system of *N. benthamiana* was used to verify the location of TaPYL5 at the cellular level, as previously described [60,61]. Briefly, protoplasts were firstly obtained from young leaves of *N. benthamiana* [60]. They were then transformed using the above expression cassettes containing TaPYL5-GFP and a modified PEG-mediated transformation protocol [61]. After 24 h of infiltration between *A. tumefaciens* transformants and explant cells, GFP signals initiated in protoplasts transformed by the fusion were detected under a fluorescent microscope.

### 4.3. Expression Analysis of TaPYL5 upon Drought and Characterization of Cis-Acting Regulatory Elements in the TaPYL5 Promoter

The expression patterns of *TaPYL5* in response to drought stress were studied using seedlings of Shimai 26, a drought-tolerant cultivar cultured in a modified MS solution supplemented with an osmotic-regulatory chemical (i.e., PEG-6000). Seeds were germinated on a regular basis, and seedlings were cultured in a standard MS solution, as previously described [62]. They were subjected to a simulated drought treatment at the second-leaf stage by growing in a modified MS solution supplemented with various concentrations of PEG-6000, including 0, 1, 5, 10, and 15% (*w/v*), corresponding to osmotic potentials of 0, −0.35, −0.81, −1.08, and −1.27 mPa, respectively. Root and leaf samples were collected over a 27-h period of the treatments, with time points of 1, 3, 9, and 27 h during drought conditions. A normal recovery treatment was established by re-growing the drought-challenged seedlings in a standard MS solution after 27 h to define the gene response to recovered normal conditions. Root and leaf tissues were collected at 1, 3, 9, and 27 h after the recovery treatment. *TaPYL5* transcripts in samples collected during drought and recovery treatments were assessed using qRT-PCR using gene specific primers (Appendix A), as previously described [63]. *Tatubulin*, a constitutive gene of wheat, was used as an internal reference to normalize the target transcripts during this time.

An online search tool referred to as PLACE (http://bioinformatics.psb.ugent.be/webtools/plantcare/html/ accessed on 12 January 2022) was adopted to identify the putatively critical *cis*-acting regulatory elements situated in the *TaPYL5* promoter. A suite of promoter regions with truncated sizes across 2 kb upstream ATG was used to functionally characterize a set of the elements in modulating transcription efficiency on the gene under drought conditions (translation start codon). PCR amplified sequences with lengths of 378, 640, 1055, 1635, and 1938 bp that contained various elements/amounts associated with gene transcription regulation and osmotic stress response (i.e., ABRE, and recognition sites MYB) (Appendix A) were separately inserted in front of reporter *GFP* in binary vector pCAMBIA3301 [64]. Transformants of *A. tumefaciens* strain EHA105 containing different cassettes were used to transform embryos of Shimai 22, as previously described [65]. Independent transgenic lines containing each construct were germinated on a regular basis, and seedlings were grown normally in an MS nutrient solution to the third leaf stage. They were then subjected to a 6 h simulated drought treatment regime by culturing in a modified MS solution supplemented with 10% (*w/v*) of PEG-6000, an osmotic-regulatory chemical for drought induction. GUS activities in transgenic lines and controls were assessed using histochemical staining for 24 h at 37 °C using GUS staining solution, which contained the following constituents: 0.1% Triton X-Gluc, pH 7.2, and 10 mM EDTA. After several wash processes with graded ethanol series, GUS staining results in samples were documented using a digital camera.

### 4.4. Yeast Two-Hybrid and BiFC Assays to Identify Protein-Protein Interactions

Yeast two-hybrid assays were carried out to identify the partner(s) of PP2C interacted with by the TaPYL5 protein as well as any SnRK2 family member(s) acting as the downstream partner of the PP2C member. Six PP2C family genes and seven SnRK2 family genes in wheat were identified using the NCBI GenBank database. The PP2C family members subjected to assay included *TaPP2C6-3B*, *TaPP2C8-1A*, *TaPP2C9-3B, TaPP2C50-1D*, and *TaPP2C53-1A* (simplified to *TaPP2C6*, *TaPP2C8*, *TaPP2C9*, *TaPP2C50*, and *TaPP2C53*, respectively); the SnRK2 family genes subjected to assay included *TaSnRK2.1*-*2A*, *TaSnRK2.2-2A, TaSnRK2.3-1A*, *TaSnRK2.4-3A*, *TaSnRK2.5-2A*, *TaSnRK2.6-2A*, and *TaSnRK2.7-1A* (renamed *TaSnRK2.1-TaSnRK2.7* for simplicity). Appendix A contains information on the genes mentioned in the PP2C and SnRK2 families. During this assay, the TaPYL5 protein expressed in yeast strain was used as bait, while the proteins in the PP2C family were expressed separately as preys. To elicit the putative protein interaction process, yeast transformants expressing TaPYL5 protein and each TaPP2C member in the host strain (AH109) were co-cultured in a selecting solid medium supplemented with exogenous ABA (1 μM). Similarly, the TaPP2C member that interacted with the TaPYL5 protein identified above was used as prey, while the proteins in the SnRK2 family were used as bait separately. The above yeast two-hybrid assays were carried out in accordance with the manufacturer’s instructions [63]. Appendix A shows the gene specific primers used for amplification of *TaPYL5* and the genes mentioned in the PP2C and SnRK2 families.

BiFC assays were used to validate protein interactions between TaPYL5 and PP2C members, as well as the PP2C protein and SnRK2 family members. The ORFs of target genes, including *TaPYL5* and members of the PP2C and SnRK2 families, were amplified by RT-PCR using gene specific primers (Appendix A) and integrated in frame with fragments of the reporter gene *YFP* (yellow fluorescent protein encoding gene) at the C- or N-terminus, as previously described [66]. *TaPYL5* and *TaPP2C2* expression cassettes, as well as *TaPP2C* and *TaSnRK2* expression cassettes, were transiently co-transformed onto *N. benthamiana* leaves using the *Agrobacterium* (stain EHA105)-mediated infiltration method [65]. Following the manufacturer’s recommendation, YFP signals in transgenic leaves were detected after 2 d of infiltration using a fluorescent microscope (Olympus FV10-ASW, Tokyo, Japan). Parallel to the detection of YFP signals, DAPI (Solarbio, Beijing, China) staining was used as a positive control to demonstrate nucleus targeting, as previously described [67].

### 4.5. Assessment for Growth Traits of Transgenic Lines

Transgene analysis on *TaPYL5* and genes encoding its partners (*TaPP2C53*, *TaSnRK2.1*, and *TaABI1*) was conducted to evaluate their roles in mediating plant drought response. *TaPYL5*, *TaPP2C53*, *TaSnRK2.1*, and *TaABI1* ORFs were amplified in both sense and anti-sense orientations using RT-PCR and gene specific primers (Appendix A). They were then separately inserted into the *Nco*I*/Bast*EII sites of the binary vector pCAMBIA3301, which was controlled by the CaMV35S promoter. Transformants of *A. tumefaciens* strain EHA105 harboring the above expression cassettes were subjected to transformation onto Shimai 22 wheat as described previously [66]. For each gene, a set of five to six transgenic lines in either sense or antisense orientation was created. qRT-PCR with gene specific primers (Appendix A) was used to assess the expression levels of the target gene in transgenic lines. To address gene functions in modulating plant drought response, two lines were selected at the T3 generation for each gene with more target transcripts and another two with lower expression levels on target genes, with one copy insertion each at the genome level. Briefly, seeds from transgenic lines and the WT were germinated on a regular basis, and seedlings were cultured in plastic pots filled with a mix of vermiculate and fertile soil (1:1, *w/w*) under normal growth conditions supplied by tap water daily to maintain 70–75% relative soil moisture. They were then subjected to drought treatment at the second-leaf stage by controlling the water supply to maintain 55–60% of relative soil moisture detected by a soil water potential metre (TRS-IIN, Shanghai, China). Growth traits in transgenic lines, including phenotypes, biomass in aerial tissues and roots, and root volumes, were evaluated three weeks after treatments. The phenotypes were documented using digital camera images, while the biomass of plant tissues and the volumes of root tissues were obtained using conventional assay methods.

### 4.6. Determination of Physiological Parameters Associated with Osmotic Stress Responses in Transgenic Lines

After normal growth and drought treatment, a subset of the physiological indices associated with plant osmotic stress responses was measured in upper expanded leaves of the transgenic wheat lines. The SCR, WLR, osmolytes contents, and ROS homeostasis-related parameters were all evaluated. The SCR values were calculated by measuring stomata aperture on the epidermis at the indicated time points (0, 0.5, 1, and 2 h) during a 2 h regime of simulated drought treatment for the third leaf stage seedlings, as previously described [67]. The WLR values were calculated based on decreased fresh weights in upper-expanded leaves at the indicated time points (0.25, 0.5, 1, 2, and 3 h) during a water deficit stressor initiated by putting on a clean bench in comparison to that assessed at time point 0 h (prior to treatment), as previously described [68]. The contents of two types of osmolytes, including proline and soluble sugar, were determined using methods described previously [67]. Among the ROS-associated parameters, antioxidant enzyme activities of SOD, catalase CAT, and POD), and MDA in transgenic and the WT leaves were determined using methods described previously [69].

### 4.7. Transcriptome Analysis

RNA-seq analysis was carried out using the 24 h drought (10% PEG, *w/v*)-stressed *TaPYL5* transgenic line (Sen 2) and the WT roots as samples. Briefly, the total RNA in root tissues of Sen 2 and WT was extracted after drought treatment using the TRIzol reagent (Invitrogen, Waltham, MA, USA). They were then used to construct strand-specific RNA-seq libraries in triplicates using the previously described method [63]. The transcripts in the libraries were sequenced on an Illumina HiSeq 2500 sequencing platform (Illumina, San Diego, CA, USA). Raw reads generated from the libraries were further processed using Trimmomatic after adaptors were removed from the reads and the reads were classified as low-quality [70]. The reads generated from the transgenic line (Sen 2) and the WT were then subjected to systematical sequence alignment analysis by searching against the *T. aestivum* reference transcript database (Novogene Co., Ltd., Beijing, China). Differentially expressed genes were identified in drought-stressed Sen 2 roots using a two-fold modified transcript abundance compared to the WT, calculated by the edger program, which effectively compares read occurrence frequencies between two group samples [71]. During this time, raw *P* values were calculated with a false discovery rate (FDR) of less than 0.05, as previously suggested [72]. Plant MetGenMap, a web-based tool (http://bioinfo.bti.cornell.edu/cgi-bin/MetGenMAP/home.cgi accessed on 10 May 2022), was adopted to characterize the pathways of KEGG for the DE genes identified in the transgenic line (Sen 2), using the pearl module program CPAN [73].

A set of randomly chosen DEGs, including five with upregulated expression patterns and five with downregulated expression patterns, were subjected to qRT-PCR analysis to validate the transcriptome analysis results. The five upregulated genes were: *TaPT5*, a phosphate transporter 5-like gene (*TraesCS3D02G439300*); *TaLD2*, a leucoanthocyanidin dioxygenase 2-like gene (*TraesCS6D02G004300*); *TaWRKY9*, a transcription factor WRKY9 gene (*TraesCS6A02G326500*); *TaNPP*, a purple acid phosphatase gene (*TraesCS4B02G348600*); and *TaF3H*, a flavanone 3-hydroxylase (*TraesCS2A02G493500*) (Appendix A); *TaWAT1*, a WAT1-related protein gene (*TraesCS3B02G006900*); *TaADH2D*, an alcohol dehydrogenase ADH2D gene (*TraesCS4A02G202300*); TaSADP, an S-adenosylmethionine decarboxylase proenzyme gene (*TraesCS6D02G202500*); *TaCBF*, an AP2 domain CBF protein gene (*TraesCS5D02G063200*) (Appendix A). qRT-PCR was used to detect the *TaPYL5* transcripts as previously described, using gene specific primers (Appendix A) and *Tatubulin* as an internal standard to normalize the target transcripts.

### 4.8. Statistical Analysis

Gene transcripts, growth traits, plant biomass, and osmotic stress-related indices were all averaged from triplicate data. The Statistical Analysis System software (SAS Corporation, Cory, NC, USA) was used to calculate standard errors for averages and statistical significance analyses for the traits studied. The significance between treatments was compared using Student’s *t*-test with the least significant difference (LSD) assessed at the level of *p* < 0.05.

## 5. Conclusions

TaPYL5 shares conserved domains with plant ABR proteins and is targeted to the stomata, plasma membrane, and nucleus following an ER selection. *TaPYL5* expression is drought-sensitive, and its transcription is determined significantly by the drought-responsive *cis*-acting regulatory elements located in the gene promoter. TaPYL5 and its two downstream partners, TaPYL5/TaPP2C53/TaSnRK2.1, were constituted to be the ABA core signaling module in yeast two-hybrid and BiFC assays. Wheat transgene analysis revealed that *TaPYL5* and *TaSnRK2.1* positively regulate plant drought tolerance while *TaPP2C53* negatively regulates it by modulating stress-related physiological indices such as stomata movement, osmolytes biosynthesis, and cellular ROS homeostasis. TaABI1 interacts with TaSnRK2.1 specifically and negatively regulates plant adaptation to drought stress, implying that it is a downstream component of the *TaPYL5*-involved signaling pathway. *TaPYL5* plays a significant role in global gene transcription regulation, with a focus on three GO terms that lead to plant physiological adaptation to drought conditions. *TaPYL5* and its downstream partners are potential targets for molecular breeding for drought-tolerant cultivars in wheat.

## Figures and Tables

**Figure 1 ijms-24-07969-f001:**
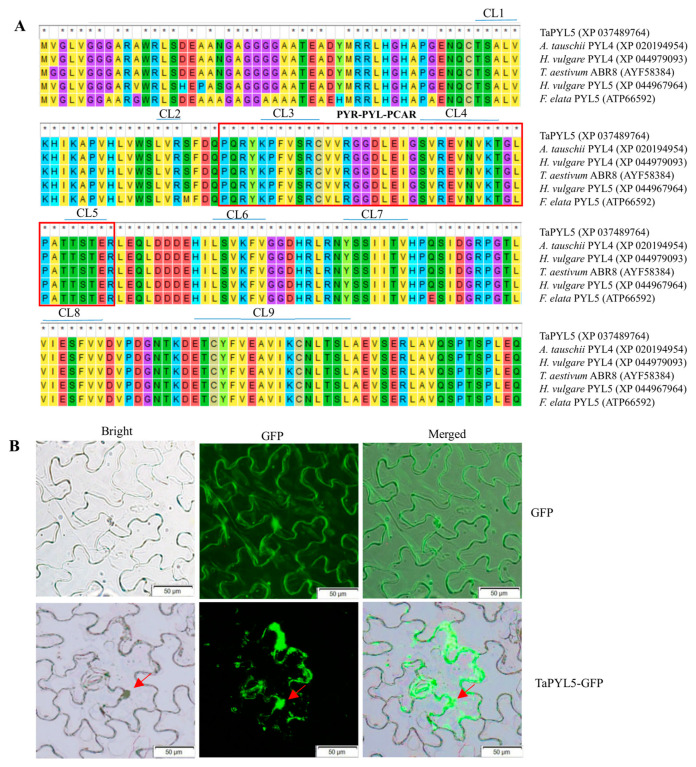
Characterization of the TaPYL5 protein. (**A**) ClustalW alignment results among the TaPYL5 protein and its counterparts in plant species. CL1 to CL9: nine conserved motifs specified by PYL proteins. PYR and PYL_PCAR: conserved domains harbored in ABA receptors. Symbol * stands for identical amino acid shared by TaPYL5 and its homologous proteins. (**B**) GFP signals in transgenic epidermal cells harboring fusion *TaPYL5-GFP* detected under fluorescent microscope. Two typical epidermal cells are round highlighted, and the arrow points to the nucleus.

**Figure 2 ijms-24-07969-f002:**
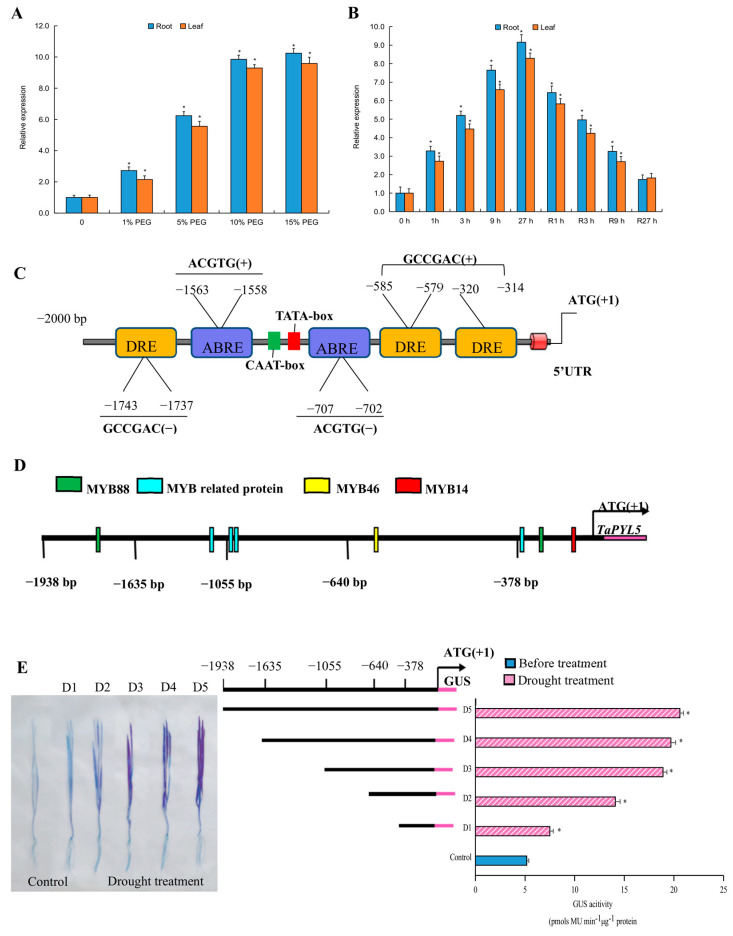
Expression patterns of *TaPYL5* and histochemical GUS staining of leaf samples integrated by *TaPYL5* promoter regions. (**A**) Expression patterns of *TaPYL5* in roots. (**B**) Expression patterns of *TaPYL5* in leaves. 1 h, 3 h, 9 h, and 27 h, time points after drought treatment. R1 h, R3 h, R9 h, and R27 h, time points during normal recovery treatment for the wheat seedlings after 27 h regime of drought stress. 0 h, time point prior to drought treatment. (**C**) Diagram showing positions of the *cis*-acting regulatory element situated in *TaPYL5* promoter. (**D**) Conserved MYB motifs. (**E**) Results for histochemical GUS staining and GUS activities. *: significant difference from the Control (*p* < 0.05).

**Figure 3 ijms-24-07969-f003:**
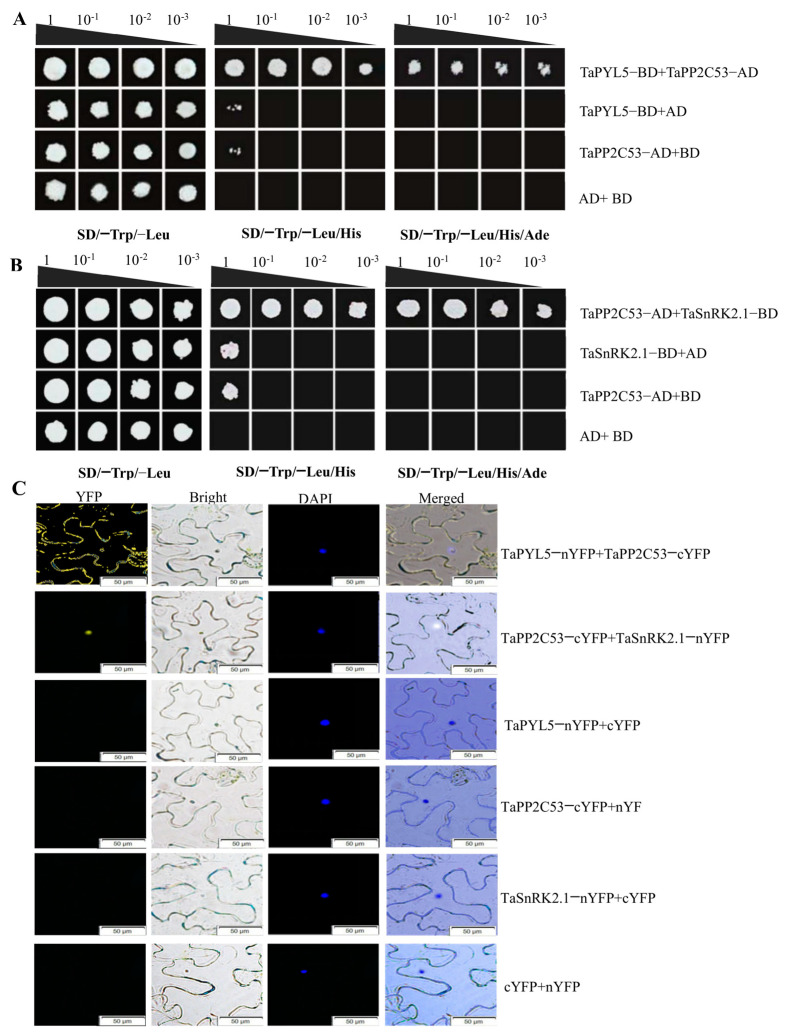
Results of yeast two-hybridization and BiFC assays on *TaPYL5* and its downstream partners. (**A**) Results for protein-protein interaction between *TaPYL5* and *TaPP2C53* detected by yeast two-hybrid assay. (**B**) Results for protein-protein interaction between *TaPP2C53* and *TaSnRK2.1* by yeast two-hybrid assay. *TaPP2C53*, a member of the A clade PP2C family in wheat. *TaSnRK2.1*, a member of the SnRK2 family in *T. aestivum.* (**C**) Results between *TaPYL5* and proteins *TaPP2C53* and *TaSnRK2.1* by BiFC assay.

**Figure 4 ijms-24-07969-f004:**
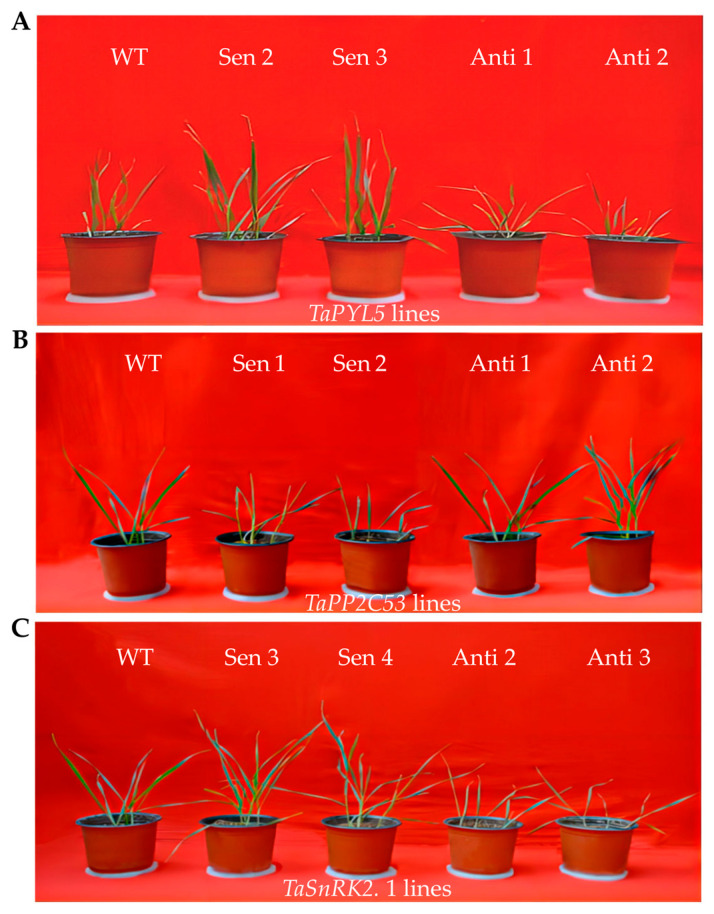
Phenotypes of *TaPYL5*, *TaPP2C53* and *TaSnRK2.1* transgenic wheat lines under drought treatment. (**A**) Phenotypes of *TaPYL5* lines. (**B**) Phenotypes of *TaPP2C53* lines. (**C**) Phenotypes of *TaSnRK2.1* lines.

**Figure 5 ijms-24-07969-f005:**
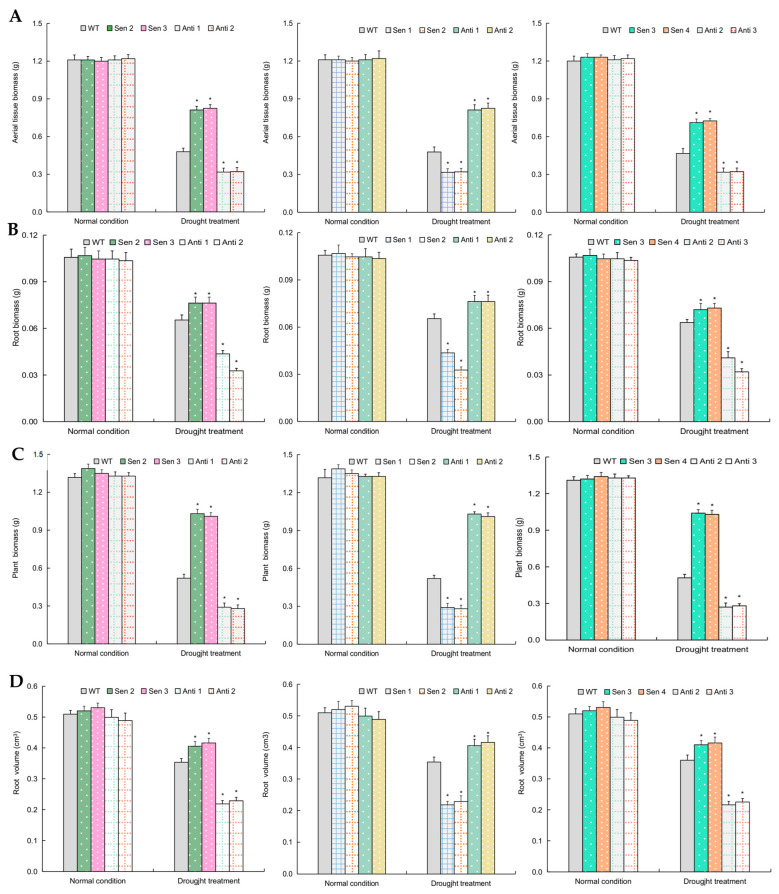
Growth traits of *TaPYL5*, *TaPP2C53* and *TaSnRK2.1* transgenic lines under normal condition and drought treatment. (**A**) Aerial tissue biomass of *TaPYL5*, *TaPP2C53* and *TaSnRK2.1* lines. (**B**) Root biomass of *TaPYL5*, *TaPP2C53* and *TaSnRK2.1* lines. (**B**) Plant biomass of *TaPYL5*, *TaPP2C53* and *TaSnRK2.1* lines. (**D**) Root biomass of *TaPYL5*, *TaPP2C53* and *TaSnRK2.1* lines. In (**A**–**D**), data shown are average plus standard error. *: significant difference from the WT (*p* < 0.05).

**Figure 6 ijms-24-07969-f006:**
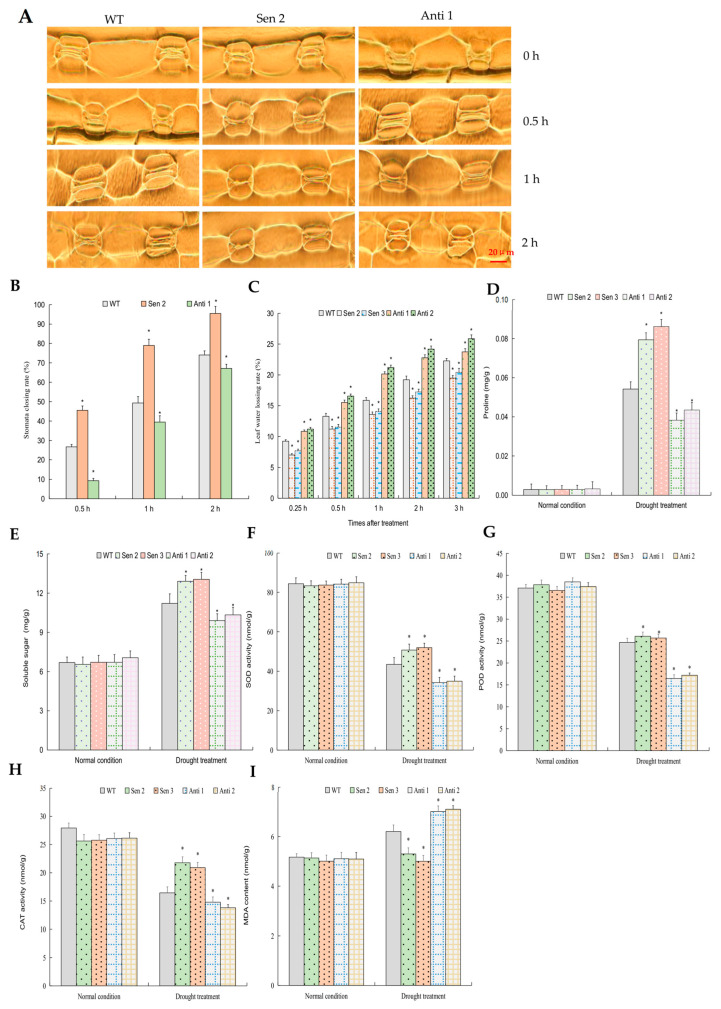
Stomata behavior and stress-related physiological indices in *TaPYL5* transgenic wheat lines under drought condition. (**A**) Stomata behaviors. (**B**) Stomata closing rates; (**C**) Leaf water closing rates. (**D**) Proline contents. (**E**), soluble sugar contents. (**F**) SOD activities. (**G**) POD activities. (**H**) CAT activities. (**I**) MDA content. In (**B**,**C**), data shown are those relative to 0 h. In (**D**–**I**), data shown are averages. *: significant difference from the WT (*p* < 0.05).

**Figure 7 ijms-24-07969-f007:**
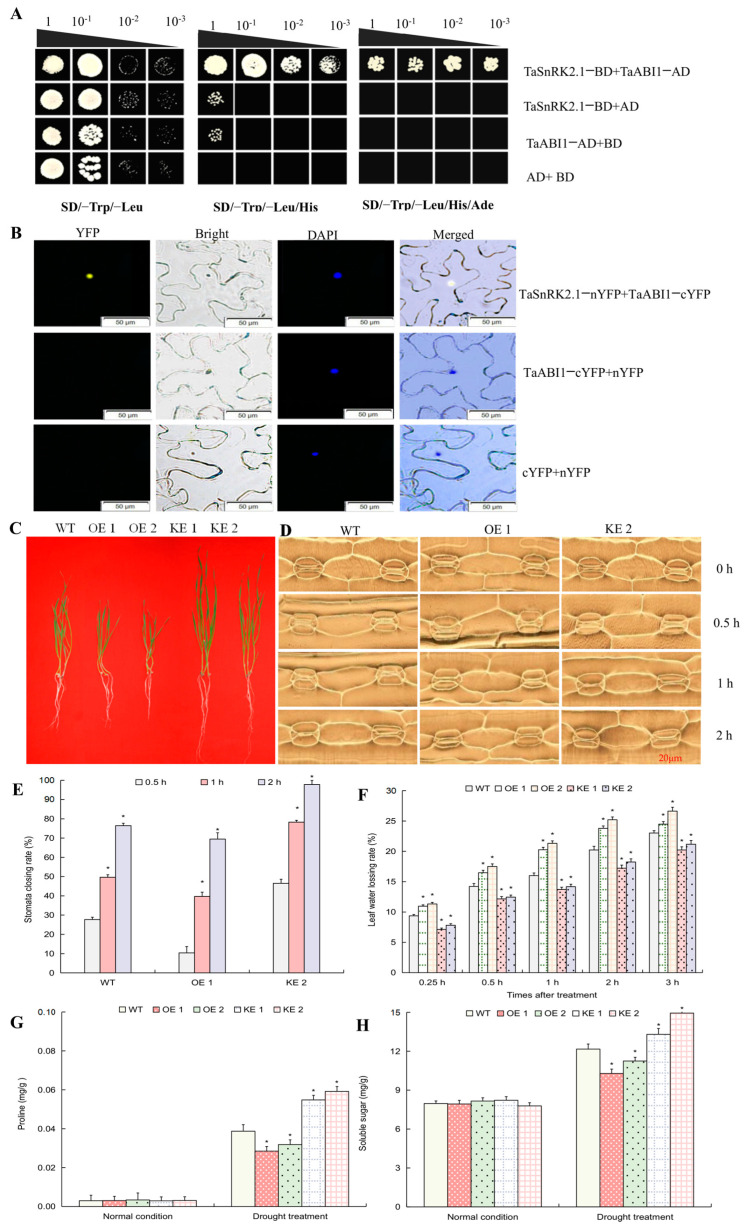
Transcriptional activation results of TaSnRK2.1 on *TaABI1.* (**A**) Yeast two-hybrid result between TaABI1 and TaSnRK2.1. (**B**) BiFC verification on interaction between TaABI1 and TaSnRK2.1. (**C**) Phenotypic and stomatal characteristics of *TaABI1* lines under drought condition. (**D**) Stomata properties of *TaABI1* lines. (**E**) Stomata closing rates of *TaABI1* lines. (**F**) Leaf water losing rates of *TaABI1* lines. (**G**) Proline contents of *TaABI1* lines. (**H**) Soluble sugar contents of *TaABI1* lines. Error bars represent standard errors. *: significant differences between the transgenic lines and WT calculated by one way ANOVA (*p* ≤ with significance level of 0.05).

**Figure 8 ijms-24-07969-f008:**
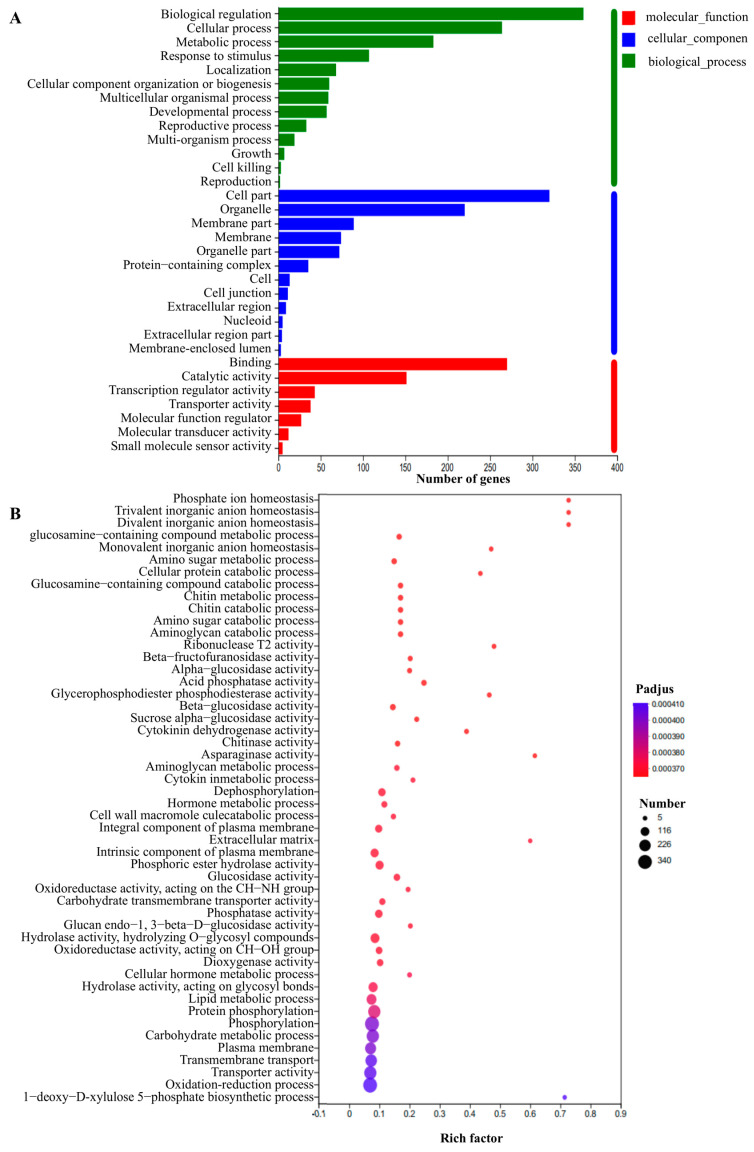
Characterization of the differently expressed genes in the drought-challenged *TaPYL5* transgenic wheat lines. (**A**) GO terms that are overrepresented by the upregulated DE genes. (**B**) biochemical pathways are enriched by the upregulated DEGs.

**Figure 9 ijms-24-07969-f009:**
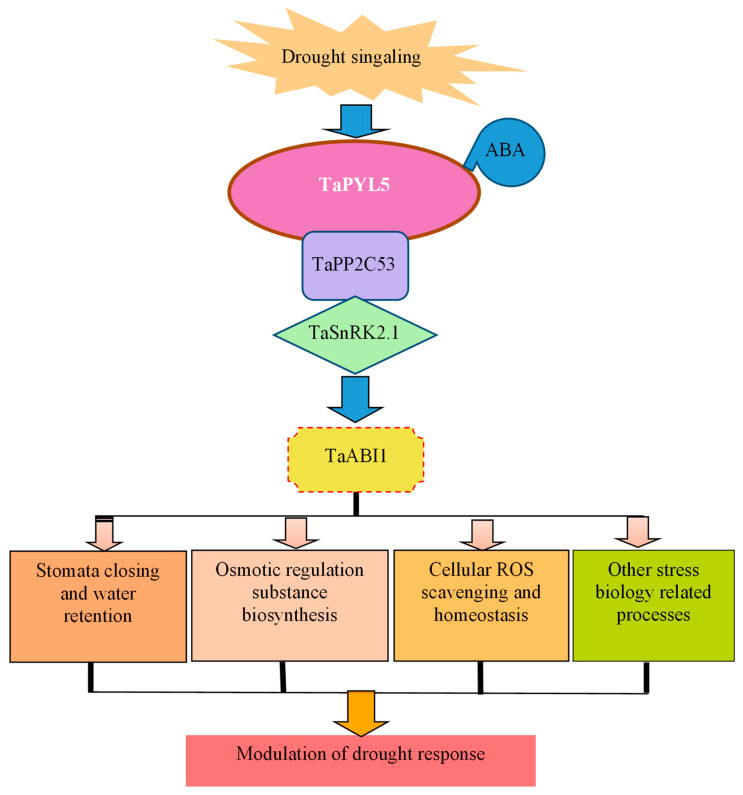
A working model defining *TaPYL5*-mediated plant adaptation to drought stress.

## Data Availability

All data generated or analyzed during this study are included in this article.

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
