# Peer review of "Wheat ABA Receptor TaPYL5 Constitutes a Signaling Module with Its Downstream Partners TaPP2C53/TaSnRK2.1/TaABI1 to Modulate Plant Drought Response"

_ijms, 2023, doi:10.3390/ijms24097969_

Round 1

Reviewer 1 Report

The study of the factors regulating plant resistance to stress is an important and topical study. In this paper, we studied the ABA dependent component, which plays an important role in the implementation of defense mechanisms during drought.

The work was done to a good standard. Methods correspond to the goals and objectives of the work. During the reading of the work, a question arose. In this paper, we are talking about ABA and the drought-sensitive component. And how will this whole system work with the use of an ABA inhibitor. since it is the ABA inhibitor that will confirm that all system operation is interrupted by ABA.

comments in the attached file.

The English language of the manuscript needs correction.

Author Response

Dear the editor and reviewers,

We are very appreciated your comments and professional suggestions on our manuscript. Based on the kind advice of you and the reviewers, we have carefully revised our manuscript. We are grateful to the reviewers for their critical and valuable remarks that are constructive and helpful for us to improve our manuscript.

We hope that you and the reviewers will be satisfied with our careful revisions and that the manuscript would be currently acceptable for publication in the International Journal of Molecular Sciences.

A point-to-point response to the reviewers’ comments is attached as below.

Thank you again for considering our manuscript. We are looking forward to hearing from you soon.

Yours sincerely,

Kai Xiao, Ph.D.

April 16, 2023

Hebei Agricultural University

State Key Laboratory of North China Crop Improvement and Regulation

No. 2596 Lekai South Street, Baoding, Hebei 071000, China

Reviewer 1 comments and our responses

Comments: The English language of the manuscript needs correction.

The study of the factors regulating plant resistance to stress is an important and topical study. In this paper, we studied the ABA dependent component, which plays an important role in the implementation of defense mechanisms during drought. The work was done to a good standard. Methods correspond to the goals and objectives of the work. During the reading of the work, a question arose. In this paper, we are talking about ABA and the drought-sensitive component. And how will this whole system work with the use of an ABA inhibitor. Since it is the ABA inhibitor that will confirm that all system operation is interrupted by ABA.

Response: We herein send our gratitude for the valuable suggestion and remarks. (1) According to the suggestion, we have entrusted an English language company (Ningbo Freescience In formation Technology co. Ltd) to polish the whole manuscript to avoid the grammatical or typical errors throughout the manuscript, by which to improve the English language of the manuscript. (2) As to the issue raised at the end of the paragraph “And how will this whole system work with the use of an ABA inhibitor. Since it is the ABA inhibitor that will confirm that all system operation is interrupted by ABA”, we will perform an experiment in future to investigate the functional mechanism of our established TaPYL5-involved signaling module (TaPYL5/TaPP2C53/TaSnRK2.1/TaABI1) using ABA inhibitors, such as fluridone, to confirm all system operation once is interrupted by ABA initiated enhanced level upon drought stress. Thanks again for this valuable comment that help us greatly for further defining mechanisms of the TaPYL5-involved signaling pathway in regulating plant drought response.

Comments shown in the attached file.

Comment 1: Line 28 (in original version): reactive oxygen species (ROS) homeostasis

Response: Thanks a lot for the valuable suggestion. As suggested, the phase “reactive oxygen species (ROS) homeostasis” has been corrected to “antioxidant enzyme function” (lines 36-37 in revision).

Comment 2: Line 30: Critical regulator

Response: Thanks for the valuable suggestion. As suggested, the phrase “critical regulator” has been corrected to “one of the most important regulators” (lines 38-39 in revision).

Comment 3: Line 33: osmotic stress-related indices

Response: Thanks for the valuable suggestion. As suggested, the phrase “osmotic stress-related indices” has been corrected to “stomata property; osmolyte content; ROS homeostasis” (line 42 in revision)

Comment 4: Line 53-59: are there any articles devoted specifically to the effect of pure ABA (not under stress conditions) on the components of the signaling pathway?

Response: We are very appreciated for this comment. As suggested, we have supplemented the related descriptions at end of the highlighted section as follows: “These findings suggested that although the signaling components constituting the ABA signaling pathway are inactive under non-stress conditions [27], they are activated upon drought and contribute greatly to the signaling transduction as well as plant stress response” (lines 79-82 in revision).

Comment 5: Line 124: core rganelle

Response: We are appreciated for this comment. The phrase “core rganelle” wrongly spelled has been corrected to “core organelle” (line 155 in revision).

Comment 6: Line 428: reactive oxygen species have not been studied given the activity of enzymes

Response: We are grateful for this comment. As suggested, the phrase “…the malondialdehyde (MDA), and accumulative ROS in transgenic and WT leaves” has been corrected to “…MDA in transgenic and WT leaves” (line 537 in revision).

Comment 7: Figure 4: Might be worth splitting these up. makes it difficult to comprehend.

Response: We are appreciated for this suggestion. As suggested, for clear demonstration of the graphs in Figure 4, we have separated this figure into two figures, namely, Figure 4 (Phenotypes of TaPYL5, TaPP2C53 and TaSnRK2.1 transgenic wheat lines under drought treatment) and Figure 5 (Growth traits of TaPYL5, TaPP2C53 and TaSnRK2.1 transgenic lines under normal condition and drought treatment) in revision. Our modification on it has thus improved the quality of graphs.  

Comment 8: Figure 5: The drawings are very tightly arranged, it is difficult to perceive them.

Response: We are very appreciated for the comment. For clear demonstration of drawings mentioned that were tightly arranged (Figure 6 in revision, original Figure 5), we have enlarged the single graphs in this figure, namely graphs 5B-5I (Figure 6 in revision), to make sure that the graphs in this figure are easily perceived.

Reviewer 2 Report

This paper deals with the investigation of the molecular functions of the wheat ABA receptor TaPYL5. It seems that TaPYL5 is an important factor that constitutes a signaling cascade related to drought stress response. Most of the presented results are solid. I have a few minor points.

1) It has been reported that TaPYL4 is an important regulatory factor underlying drought stress response. Please discuss the comparison between TaPYL4 and TaPYL5. Your laboratory published the paper "TaPYL4, an ABA receptor gene of wheat, positively regulates plant drought adaptation through modulating the osmotic stress-associated processes". In addition, "Tuning water-use efficiency and drought tolerance in wheat using abscisic acid receptors. Nat Plant. 2019;5:153–9." is also one of the important paper regarding TaPYL4. 

2) In Fig 4 and Fig 5, due to low resolution, I could not catch the characters of the vertical axis in the graphs.

3) In the attached PDF manuscript, the year number is "2021", although this comment might not be directly for the authors.  

Author Response

Dear the editor and reviewers,

We are very appreciated for your comments and professional remarks. Based on the kind suggestions from the reviewers, we have carefully revised our manuscript. We are grateful to the two reviewers for their critical comments that are constructive and helpful for us to improve our manuscript.

We hope that you and the reviewers will be satisfied with our careful revisions and that the manuscript would be currently acceptable for publication in the International Journal of Molecular Sciences.

A point-to-point response to the reviewers’ comments is attached as below.

Thank you again for considering our manuscript. We are looking forward to hearing from you soon.

Yours sincerely,

Kai Xiao, Ph.D.

April 16, 2023

Hebei Agricultural University

State Key Laboratory of North China Crop Improvement and Regulation

No. 2596 Lekai South Street, Baoding, Hebei 071000, China

Reviewer 2  Comments and our responses

This paper deals with the investigation of the molecular functions of the wheat ABA receptor TaPYL5. It seems that TaPYL5 is an important factor that constitutes a signaling cascade related to drought stress response. Most of the presented results are solid. I have a few minor points.

Comment 1: 1) It has been reported that TaPYL4 is an important regulatory factor underlying drought  stress response. Please discuss the comparison between TaPYL4 and TaPYL5. Your laboratory published the paper “TaPYL4, an ABA receptor gene of wheat, positively regulates plant drought adaptation through modulating the osmotic stress-associated processes”. In addition, “Tuning water-use efficiency and drought tolerance in wheat using abscisic acid receptor. Nat Plant. 2019; 5: 153-9.” Is also one of the important paper regarding TaPYL4.

Response: We are appreciated for the valuable suggestion and remarks. As suggested, we have discussed TaPYL5, the ABA receptor gene characterized in this study, in comparison with another member of the wheat ABA receptor gene TaPYL4 that was reported in our previous study: “TaPYL4, an ABA receptor gene of wheat, positively regulates plant drought adaptation through modulating the osmotic stress-associated processes” (BMC Plant Biology, 2022, 22: 423) and in the literature: “tuning water-use efficiency and drought tolerance in wheat using abscisic acid receptor.” (Nature Plants, 2019; 5: 153-159). The related descriptions supplemented in “Discussion” section are as follows:

(1) In section 3.2 of Discussion: “3.2. TaPYL5 Is Involved in Establishing an ABA Signaling Module with Distinct PP2C and SnRK2 Members to Contribute Plant Drought Response”, we have supplemental descriptions to illustrate our research on TaPYL4-involved constitution of an ABA core signaling module: “Our previous investigation on TaPYL4, a member of the ABA receptor family, indicating that it is involved in constitution of an ABA core signaling module with PP2C member TaPP2C2 and SnRK2 member TaSnRK2.1 [37]” (lines 284-287 in revision).

(2) In section 3.3 of Discussion: “3.3. TaPYL5 and Genes Encoding Its Partners Contribute Largely to Plant Drought Response”, we have supplemental descriptions to illustrate the functions of TaPYL4 in mediating plant drought tolerance: “In wheat, previous investigations on TaPYL4 have indicated that it acts as one of the critical regulators in positively regulating plant drought tolerance [37, 49]. The transgenic wheat lines with TaPYL4 overexpression displayed increased plant ABA sensitivity, lowered water consumption during plant's lifetime by reducing transpiration, increased photosynthetic activity, and improved water-use efficiency under water deficit conditions [49].” (lines 317-322 in revision).

(3) In section 3.3 of Discussion: “3.3. TaPYL5 and Genes Encoding Its Partners Contribute Largely to Plant Drought Response”, we have supplemental descriptions to illustrate the physiological process associated with TaPYL4-mediated plant drought tolerance: “Our investigation on TaPYL4 revealed that this wheat ABA receptor gene-mediated plant drought response is associated with its role in transcriptionally regulating TaPIN9, a PIN-FORMED gene modulating cellular auxin translocation. The modified expression of this PIN-FORMED member improves establishment of root system architecture (RSA) underlying TaPYL4 regulation to contribute to plant drought adaptation [37].” (lines 333-338 in revision).

Comment 2: 2) In Fig. 4 and Fig. 5, due to low resolution, I could not catch the characters of the vertical axis in the graphs.

Response: We are appreciated for the valuable suggestions. As suggested, we have separated original Figure 4 into two figures, namely Figure 4 (Phenotypes of TaPYL5, TaPP2C53 and TaSnRK2.1 transgenic wheat lines under drought treatment) and Figure 5 (Growth traits of TaPYL5, TaPP2C53 and TaSnRK2.1 transgenic lines under normal condition and drought treatment) in revision. In addition, we have enlarged the graphs in Figure 5 (Figure 6, (B) to (I) in revision) by which to easily catch the characters of the vertical axis in the graphs. We hope that our modification on them could effectively improve the quality of graphs in these figures.

Comment 3: 3) In the attached PDF manuscripts, the year number is “2021”, although this comment might not be directly for the authors.

Response: We are again grateful for your hard work on evaluating our manuscript and for the valuable and constructive suggestions that help us improve the manuscript quality.
